# Muonium reaction in MgO: A showcase for the final steps of ion implantation

Rui C. Vilão[1]⋆, Ali Roonkiani[1], Apostolos G. Marinopoulos[1], Helena V. Alberto[1],
João M. Gil[1], Ricardo B. L. Vieira[1], Robert Scheuermann[2] and Alois Weidinger[3]

**1** University of Coimbra, CFisUC, Department of Physics, P-3004-516 Coimbra, Portugal
**2** Laboratory for Muon-Spin Spectroscopy, Paul Scherrer Institut,
5232 Villigen PSI, Switzerland
**3** Helmholtz-Zentrum Berlin für Materialien und Energie,
Department ASPIN, 14109 Berlin, Germany

⋆ ruivilao@uc.pt

## Abstract

We present an in-depth investigation of the implantation of positive muons in magnesium oxide (MgO). Muonium, the positive muon plus an electron is an analogue of the hydrogen atom. This study describes the final stage of the implantation process, from muon diffusion over the potential barrier and the stopping by an inelastic reaction to the final embedding of the muon into the lattice structure. A special aspect is a relatively long-lived intermediate configuration which lasts for several hundred nanoseconds or more and is accessible to muon spin spectroscopy. The model presented here provides a framework for the analysis of the general case of ion implantation.

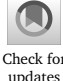

# 1  Introduction

The stopping process of ions in solids is a topic of great importance for a wide range of scientific and technological subjects, including nuclear industry [1], high-energy density physics [2], inertial confinement fusion [3], hadron-therapy in medicine [4], photovoltaic technology for space applications [5], microelectronics [6], or in ion beam technologies [7]. Although the stopping process is rather well understood for the high-energy region, very little is known about the final thermalization process [8–12]. The reason for this is that conventional experimental methods are not able to investigate this process at the atomic level. Muon spectroscopy offers the unique possibility that the final steps can be observed directly on the implanted particle and thus detailed information about these final steps can be obtained. A particularly important case is the stopping of protons in matter [13–16].

In this work we investigate the final steps in the implantation of positive muons in the dielectric oxide MgO, using muon spin spectroscopy ($\mu$SR). $\mu$SR is an established experimental method in materials research [17, 18]. Despite their leptonic nature, positive muons can be considered in condensed-matter physics as light pseudo-isotopes of hydrogen, being only c. 9 times lighter than the proton. These particles can therefore be used to model the behaviour of protons in condensed-matter systems, also allowing to extend towards isotopic studies of hydrogen in the high-dilution limit, which is typically hard to investigate [19–21]. The use of $\mu$SR lies in the implantation of positive muons in solids with a high kinetic energy of 4 MeV (although muons moderated to keV energies are sometimes used as well [22]) and on the experimental observation of the final stable configurations, from which information about hydrogen is extracted.

The stopping process of the 4 MeV muons is typically very fast and is not easily accessible experimentally in the time-window of the $\mu$SR experiments. The details of the thermalization process are nevertheless crucial for a correct assessment of the final observed muonium configurations and their translation for the physics of hydrogen in semiconductors [23, 24]. The energy dissipation process begins with the well-known Bethe-Bloch mechanism, followed by a fast charge-exchange regime which ends up when the kinetic energy of the muon is lower than the lowest available electronic excitation (1 to a few eV, the typical order of magnitude of the bandgap energy in semiconductors and insulators) [25]. The mechanism for the dissipation of the remainder of the implanted ion kinetic energy to thermal energies remains however largely unknown [25–27]. As is well known for long, phonons play a crucial role in the trapping of the muon/proton in its final stopping site [28].

Usually, information on the embedding process of the muon in the lattice is drawn indirectly from the stable configurations finally observed in the experiment. In semiconductors and insulators and below room temperature, these reactions occur frequently on a time scale of nanoseconds to microseconds and thus fall within the time window of $\mu$SR [25, 29, 30]. In these cases, the $\mu$SR method offers a unique possibility to explicitly observe also the intermediate steps before the muon is finally incorporated in the lattice.

This possibility offered by $\mu$SR allowed to unveil details of these last stages. In particular, in past experiments in oxides it was shown that a fast-relaxing muon component corresponded to a transient configuration with a fluctuating and very small electronic spin density at the muon [30]. A barrier model has been developed for the branching from this "transition-state" to the final configurations (oxygen-bound and interstitial muonium) [31]. We have also identified a thermal spike effect due to the energy liberated during the muon stopping process and the stress release by the relaxation of the lattice [32], akin to the long known thermal spike effect thouroughly described in the ion implantation literature [33–35]. This model was subsequently and successfully applied to several oxide and semiconducting systems [36–39]. A critical discussion of this model, with a thorough review of previous literature, can be found in Refs. 24, 26 and 27.

In this work we aim at further developing the previous model by elucidating the mechanism by which muonium is finally incorporated into the host lattice. We develop and present a model whose basic assumption is that muonium is eventually stopped by a strong inelastic process leading to a temporary muonium configuration which subsequently decays into the final configurations. Because of the central role of this inelastic process, which stands at the beginning of the whole reaction, we use the names "doorway model" and "doorway state" for the corresponding model and state. It indicates that the further course of the reaction is via this entrance configuration. "Doorway model" expresses more clearly than the previously used designation "transition state model" the entrance character of the initial reactions: The passage through the "doorway" by the inelastic reaction and the formation of an initial configuration through which all other reactions proceed.

In order to develop and present this doorway model, we use magnesium oxide (MgO) as a benchmark solid and put forward experimental evidence on this last stage of thermalization. The interest in the investigation of the muonium states in MgO arose from the success of muonium studies in the clarification of the behaviour of the hydrogen impurity in semiconductors and insulators [19–21]. Magnesium oxide is a rather versatile material currently being used or in consideration for applications in magnetic tunnel junction, catalysis, photovoltaics or photonics [40–44]. The attraction of hydrogen to the MgO surface and its influence on hydrogen migration has been studied [41, 45]. The investigation of hydrogen in MgO has few limited studies in the past, due to the usual difficulty in investigating hydrogen impurity directly [41, 46, 47], although hydrogen treatments are usually performed in contemporary device building [48].

Some $\mu$SR experiments on MgO have been carried out in the past [20, 23]: atomic-like muonium has been reported both at 6 K and at room temperature with significant change in its relaxation rate [49–51]. The corresponding reduced hyperfine parameter at 300 K was reported to be 0.86246(6), corresponding to 3849.4(3) MHz [52]. $\beta$-NMR experiments in MgO also revealed a frequency shift of 100 ppm that the authors consider to be likely from a significant transient population of carriers related to the implantation itself [53].

The present paper is therefore centered in the presentation and development of the doorway model, taking these novel data on MgO as a particularly illuminating example. The paper is organized in the following way: in the next section (section 2) we present a summary of the experimental results. In section 3, we present our theoretical framework: we describe the specific properties of MgO in connection with first-principles density functional theory (DFT) calculations [54, 55], and develop our "doorway model" for the last stage of the muon thermalization and embedding in the crystal lattice. In section 4 we discuss the experimental findings at the light of the doorway model and obtain the properties of the doorway state in MgO. The main findings are summarized in section 5 and additional details of the experimental and theoretical methods are presented in the appendix.

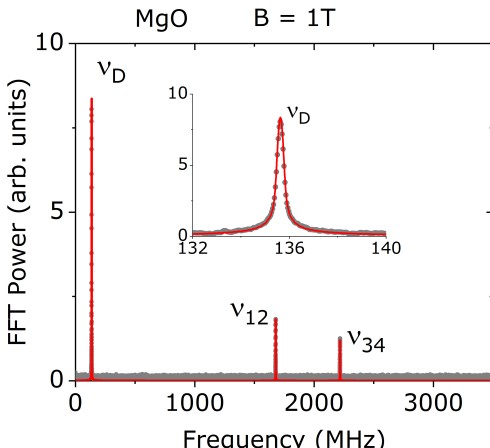

Figure 1: Fourier transform of the $\mu$SR spectrum at temperature T = 6 K and a transverse magnetic field B = 1 T. The diamagnetic-like component $\nu_D$ and the two components $\nu_{12}$ and $\nu_{34}$ of muonium are seen. The insert shows on an extended scale the broad diamagnetic-like line which is attributed to the doorway state (see text).

## 2 Experimental results

MgO is an insulator with a bandgap of 7.8 eV [56], crystallizing in the face-centered cubic (fcc) sodium chloride structure. Our sample was a commercial high-purity ($> 99.99\%$) MgO single crystal with dimensions $10 \times 10 \times 1$ mm$^3$. The orientation of the sample was such that one of the axes pointed in the beam direction (flat side of the sample), and the other two axes were perpendicular to it in the horizontal and vertical direction. More details about the sample are given in section A.1 of the appendix.

The muon spin rotation ($\mu$SR) experiment was performed at the Swiss Muon Source of the Paul Scherrer Institut in Switzerland [57] with the high-field spectrometer (HAL-9500) [58] in transverse geometry. A magnetic field $B = 1$ T was used. The magnetic field was applied parallel to the beam direction which means, considering the crystal orientation mentioned above, parallel to a cube axis of the fcc structure. The muon spin was rotated, before the implantation, from its original direction (antiparallel to the beam) towards the vertical direction to allow the transverse field measurement. Figure 1 shows the Fourier spectrum of $\mu$SR data at T = 6 K and B = 1 T.

The precession lines observed in the $\mu$SR time spectra were analyzed with either a gaussian or a lorentzian relaxation function:

$$A(t) = A_0 \exp\left(-\frac{1}{2}\sigma^2 t^2\right) \cos(2\pi \nu t + \phi) \,, \tag{1}$$

or

$$A(t) = A_0 \exp(-\lambda t) \cos(2\pi \nu t + \phi) \,, \tag{2}$$

where $A_0$, $\nu$, $\phi$, correspond to the amplitude, frequency and initial phase, and $\sigma$ and $\lambda$ to the relaxation rates, respectively. The fraction $f$ of muons forming each configuration can be determined from comparison with the maximum instrumental asymmetry $A_{\max}$ determined from a silver calibration, as $f = A/A_{\max}$. The experimental arrangement of the HAL-9500 spectrometer consisted of eight forward and eight backward detectors, arranged in rings around the muon beam. The detectors were combined to obtain a single parameter set (see section A.1 of the appendix).

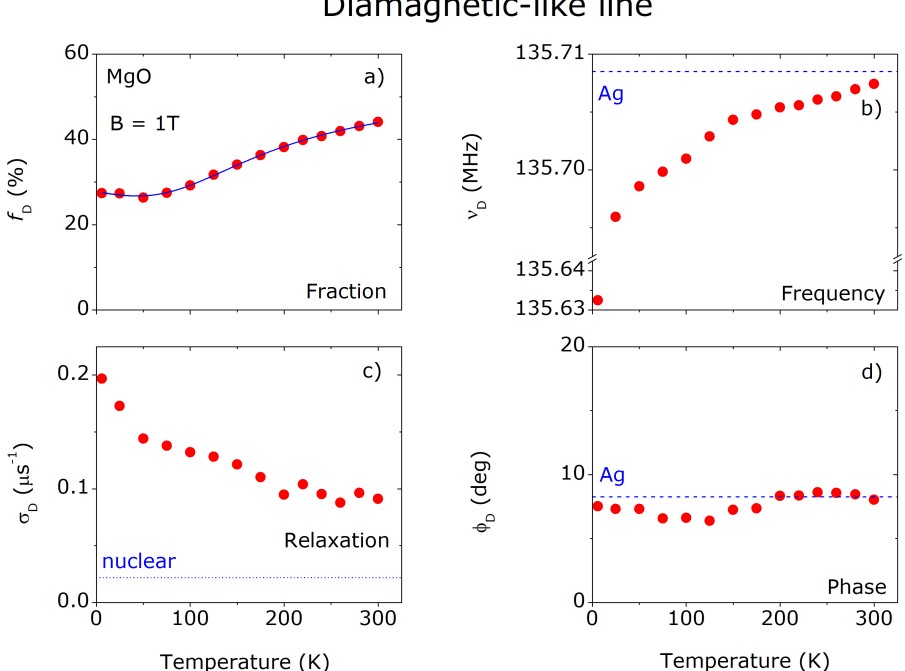

Figure 2: Temperature-dependence data (fraction $f_D$, relaxation $\sigma_D$, frequency $\nu_D$ and phase $\phi_D$) of the diamagnetic-like signal in MgO at a nominal external applied magnetic field $B = 1$ T. Frequency and phase from a calibration with Ag are indicated as dashed blue lines in Fig. 2 b), and d). The dashed blue line in Fig. 2 c) corresponds to the calculated $\sigma_D$ from nuclear magnetic moments. Note the interrupted vertical scale for the 6 K point in Fig. 2 b). The error bars are smaller than the size of the symbols.

In $\mu$SR experiments one can distinguish between two fundamentally different states, diamagnetic or paramagnetic, by the different precession frequencies or relaxation of the muon spin polarization. Diamagnetic refers to the charged muonium state $Mu^+$ or $Mu^-$, where there is no interaction between the muon and an unpaired electron. When such an interaction is present, the configuration is called paramagnetic, designated as muonium ($Mu^0$) [17, 29]. If the hyperfine interaction of a paramagnetic state is very small, the apparent precession frequency is similar to that of the diamagnetic state and is therefore termed as "diamagnetic-like".

In Fig, 1, three lines, a diamagnetic-like line $\nu_D$ and the two paramagnetic components $\nu_{12}$ and $\nu_{34}$ of muonium are seen. The line at the muon Larmor frequency $\nu_D$ (shown on an extended scale in the insert) is broader than expected for a purely diamagnetic line, indicating that some paramagnetic interaction is present. We therefore call this line diamagnetic-like. It corresponds to the fast relaxing signal discussed in the literature [26, 30, 37]. A connection of these findings with the DFT calculations will be presented in section 3.

Figures 2, 3 and 4 summarize the experimental results. In Fig. 2 the results of the diamagnetic-like signal at B = 1 T are displayed as a function of temperature. The diamagnetic-like fraction (Fig. 2 a)), as calculated by comparing the asymmetry of the diamagnetic signal with the maximum instrumental asymmetry obtained from a Ag calibration sample, is approximately constant (around 28%) up to about 100 K, then increases to about 42% at 300 K. The increase indicates a thermal activation of the formation of this state. The solid line is a fit as explained in section 4 (subsection 4.1).

The frequency of the diamagnetic-like signal (Fig. 2 b)) varies with temperature. The shift is very small, but significant. It is due to the paramagnetism of the electron which affects the muon spin via the hyperfine interaction.

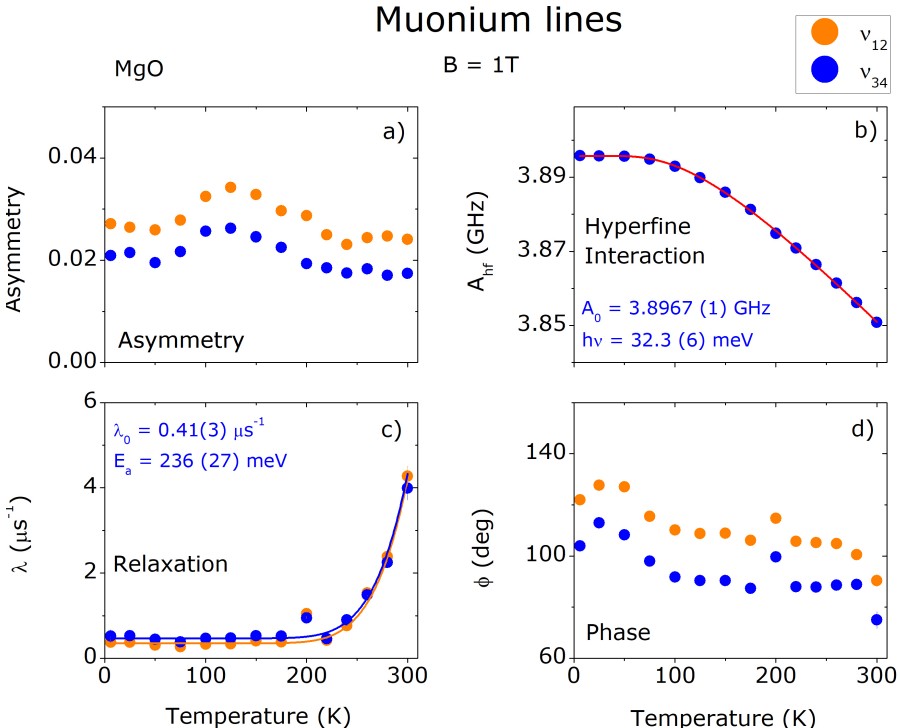

Figure 3: Temperature dependence of the two muonium lines $\nu_{12}$ and $\nu_{34}$ in MgO at B = 1 T. (a) Asymmetry of the two lines. (b) Hyperfine interaction $A_{hf}$ deduced from the two frequencies. (c) Relaxation rates $\lambda$. (d) Initial precession phases $\phi$ for each line. The solid lines in (b) and (c) are fits as explained in the text. The obtained parameters are quoted in the figure. Most error bars are smaller than the size of the symbols.

The relaxation $\sigma_D$ (Fig. 2 c)) is relatively small but clearly larger than the relaxation expected from the interaction with nuclear spins only. The nuclear interaction ("nuclear" in the figure) was calculated [59,60] for the muon bound to an oxygen in the antibonding direction (see section 3 for details) and for the interaction of the muon spin with the nuclear moments of the spin-carrying isotopes of MgO. The increased relaxation indicates interaction with an electron.

The phase (Fig. 2 d)) corresponds approximately to that of the Ag calibration. No significance is attributed to the small variation with temperature.

Figure 3 shows the data for atom-like muonium in the high field (B = 1 T) experiment. Fig. 3 a) displays the amplitudes of the $\nu_{12}$ and $\nu_{34}$ signal. The difference of the amplitude values of the two lines is due to a frequency dependent time resolution factor which affects the amplitude of the higher frequency $\nu_{34}$ more than that of the lower frequency $\nu_{12}$. The apparent peak in the amplitudes around $T = 100$ K will be discussed in connection with Fig. 4.

The hyperfine interaction $A_{hf}$ (Fig. 3 b)) was obtained from the two frequencies via the diagonalization of the corresponding Hamiltonian [29]. The red solid line is a fit with an Einstein model with a single vibrational energy $h\nu$. In this model it is assumed that the change of $A_{hf}$ is proportional to the mean square displacement $<u^2>$ of the atoms from the muon [61]. For isotropic harmonic oscillator and Bose-Einstein statistics, the corresponding formula is:

$$A_{hf} = A_0 + \frac{C}{\exp\left(\frac{h\nu}{k_B T}\right) - 1}, \tag{3}$$

where $A_0$ is the extrapolated hyperfine interaction at $T = 0$ K, $h\nu$ is the vibrational energy, $T$

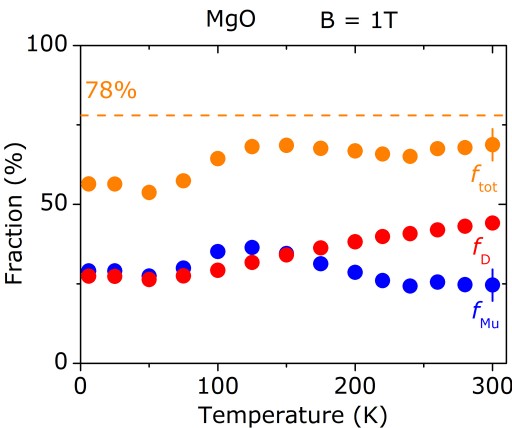

Figure 4: Temperature dependence of the fractions of muons forming the diamagnetic-like component $f_D$, the muonium component $f_{Mu}$, and the sum of the two $f_{tot}$. Also shown, as a dashed line, is the total fraction of 78%, obtained in a low field experiment (B = 1.5 mT) at T = 300 K, where no dephasing is observed.[1] The error bars at the last points of $f_D$ and $f_{tot}$ represent the uncertainty due to the time-resolution correction. Statistical errors are smaller than the size of the symbols.

is the temperature and $k_B$ is the Boltzmann constant. C is a constant fitting parameter: it is related to the coupling strength of the muonium electron with the phonons, but no details can be given here. The fit yields: $A_{hf}$ (T = 0 K) = 3896.7(1) MHz (corresponding to approximately 87% of the vacuum value) and $h\nu$ = 32.3(6) meV. The value of the vibrational energy is similar to that of an optical phonon in which atoms of the two sublattices vibrate against each other [62].

The relaxation rate $\lambda$ (Fig. 3 c)) is approximately the same for both frequencies and constant at low temperatures. Above about 250 K, the relaxation increases. We performed a fit to both relaxations with common parameters with $\lambda = \lambda_0 + \lambda_1 \exp(-E_a/k_B T)$ and obtained $\lambda_0 = 0.38(3)$ $\mu s^{-1}$ and $E_a = 236(27)$ meV. In the final fit, $\lambda_1$ was fixed to $3 \times 10^4$ $\mu s^{-1}$. It is unlikely that the increase of the relaxation with temperature is due to direct ionization of muonium, since the DFT calculations (see section 3, subsection 3.2) predict the $Mu^0$ (+/0) conversion level at 1.84 eV below the conduction band. The very fast increase of the relaxation above 200 K is very untypical for trapping. Such a fast increase has been seen in many other systems in $\mu$SR and is indicative of a conversion of one state to another with very different hyperfine interactions [29,31]. We therefore suggest that the 236 meV activation energy is associated with the conversion of the interstitial muonium into a metastable bound configuration (doorway state, see section 3, subsection 3.4). The activation for this conversion requires less energy than the ionization of muonium without a configuration change.

Figure 3 d) shows the phase of the two frequencies at the instant $t = 0$. The time zero as defined by the spectrometer electronics does not necessarily coincide with the beginning of the muon spin rotation in the sample. This circumstance complicates the evaluation of the phase shift for high frequency lines. Because of the large uncertainties we have not further analysed the phases.

Figure 4 gathers the temperature dependence of the fraction of muons forming the diamagnetic-like signal ($f_D$, from Fig. 2 a)) and of the fraction of muons $f_{Mu}$ forming the muonium state (from Fig. 3 a)), after correcting for the time-resolution effect, as explained in subsection A.1 of the appendix. The muonium fraction $f_{Mu}$ in Fig. 4 is obtained by summing the

---

[1]$\mu$SR experiment on the same MgO sample, at B = 1.5 mT and T = 260 − 360 K, performed at the GPS spectrometer at the Paul Scherrer Institut.

time-resolution corrected $f_{12}$ and $f_{34}$ Mu fractions obtained from Fig. 3 a). The intensity in the remaining two components $\nu_{23}$ and $\nu_{14}$ is negligible at this high field (B = 1 T). The total fraction $f_{\text{tot}}$, the sum of $f_D$ and $f_{\text{Mu}}$, is also shown in Fig. 4, as well as an extrapolation of the total fraction (corresponding to a fraction of 78%) obtained in a low-field experiment,[1] where no dephasing was observed. As mentioned above, the observed muonium amplitudes show a peaking slightly above 100 K. We note that the increase is parallel to the increase of diamagnetic-like fraction, suggesting that both states are formed in the same process. The decrease of the muonium fraction at temperatures above about 120 K may either be due to a decrease of the formation probability or to dephasing effects which are rather strong at these high frequencies. About 20% fraction is missing. It is attributed to muon spin polarization loss due to rapid fluctuations of the hyperfine interaction in the initial hot phase after the muon stopping.

Broadened diamagnetic lines have been observed in the past in other systems and interpreted as shallow donor muonium states (see e.g. Ref. 63). The case in MgO is not different from these previous cases and the properties of these states are similar to those observed in the present MgO case. However, in the past works, no detailed information is given on the formation process and the geometrical structure of these states. In the present case we describe the full history of this state from its formation to its decay and give the geometrical configuration adopted at the different steps along this process.

## 3 Theory

### 3.1 Ab-initio (DFT) calculations

Muonium in MgO was investigated using density functional theory (DFT) calculations with ab-initio pseudopotentials [64–66]. Exchange and correlation effects between the electrons were described by the semilocal PBE functional [67] and the screened-exchange HSE06 hybrid-functional approach [68, 69]. In these calculations, the muon particle is represented by a proton, an approximation that considers muonium to be the lighter pseudo-isotope of monatomic hydrogen. This approximation is expected to yield nearly identical ground-state properties for these species [20, 29]. Zero-point energies were explicitly treated in total energy barrier calculations to account for the different isotopic masses. A more detailed description of the DFT method can be found in subsection A.2 of the appendix.

Not only ground state but also metastable excited states were considered. They play a role as intermediate configurations in implantation and doping processes. Electron configurations are shown to demonstrate the interaction of hydrogen with the surrounding atoms. Energy profiles along the diffusion path were calculated, both for frozen lattice and also accounting for structural relaxation (relaxed lattice). The frozen lattice configuration is relevant for the fast-diffusing particle, whereas the relaxed configuration represents the situation after stopping.

### 3.2 Muonium configurations in MgO

MgO crystallizes in the face centered cubic (fcc) sodium chloride structure (Fig. 5 a)). The unit cell consists of cubes, each containing four Mg and four O atoms in the corners. Muons with a kinetic energy of 4 MeV are implanted into this structure. In the end phase of the thermalization process, the resulting neutral muonium atom ($\text{Mu}^0$) (the muon has picked up an electron during the slowing down process) diffuses through the lattice by hopping between interstitial sites. The diffusion path is indicated by the arrow in Fig. 5 a). The final stable configurations are: i) neutral muonium in the center of a cube (Fig. 5 b)) with the electron density (in yellow) concentrated on the muon, and ii) ionized muonium $\text{Mu}^+$ bonded to an



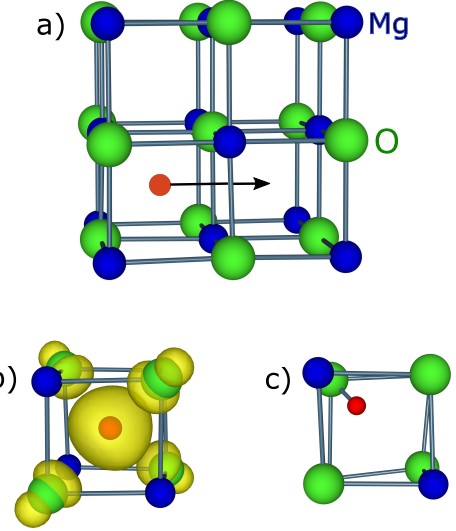

Figure 5: a) Sodium chloride structure of MgO. Also indicated is the position of the muonium (red sphere) in the center of the cube and the expected diffusion path from one cube to the next (arrow). b) Calculated (DFT) position of neutral muonium in the center of the cube (red dot) and the electron-spin distribution around the muon (yellow cloud). c) Calculated (DFT) position of the positive muonium ($Mu^+$) bound to an oxygen atom.

oxygen atom (Fig. 5 c)). As mentioned in section 2, we assign the experimentally observed muonium lines to the calculated neutral muonium state shown in Fig. 5 b). The purely ionized muonium $Mu^+$ bonded to an oxygen atom (Fig. 5 c), is contained within the observed diamagnetic-like line discussed in section 2.

The lowest-energy (ground state) site for neutral interstitial muonium is in the center of a cube. Examination of the ground-state charge density for this defect shows that the muonium electron has a strong 1s-type spherically symmetric character and it is centered at the muon site (see Fig. 5 b)). Some residual finite spin density exists also at the four neighboring oxygen atoms of the cube. This suggests that muonium polarizes to some degree its immediate environment. The calculated hyperfine interaction is dominated by an isotropic term, $A_{iso}$. The magnitude of $A_{iso}$ obtained by the hybrid HSE06 functional is approximately 91% of the vacuum value, in reasonable agreement with the experimental value for muonium (87%) (see section 2). In addition to this state, muonium forms a bound configuration where the muon is displaced from the cube center and forms a O—Mu bond in analogy to the hydroxyl O—H type bond (Fig. 5 c)). This O—Mu bound configuration was found by the DFT calculations to be thermodynamically stable solely in its positively-charged state. The (+/0) conversion level was found to lie deep in the gap, at 1.84 eV below the conduction-band edge.

## 3.3 Diffusion barrier for muonium migration

Towards the end of the thermalization process, muonium diffuses with the remaining kinetic energy from one interstitial site to the next through the square plane between the two sites (Fig. 5 a)). On symmetry grounds, the lowest energy path is through the center of this plane, which corresponds to the highest point of the energy barrier between the stable positions at the center of the cubes. The calculated energy profile and corresponding electron-densities are displayed in Fig. 6. It can be seen that muonium retains its strong atomic character for

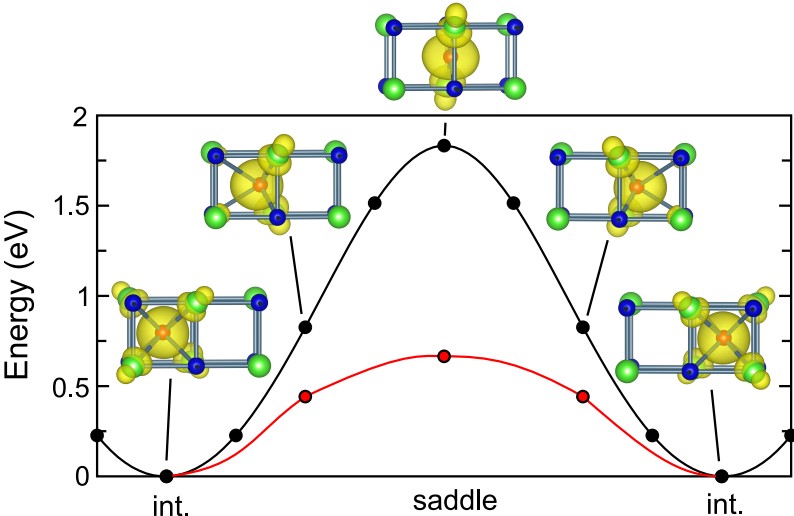

Figure 6: DFT energy profile and electron densities for intermediate muonium positions along the diffusion path of muonium from one cube center to the next through the common cube face: black line for the non-relaxed frozen lattice; red line for the relaxed lattice. The path of the muon is indicated. The points were calculated with the HSE06 functional, the lines are guides to the eye.

large portions along the path. The energy maximum occurs at the midpoint of the path when the nucleus occupies the face center position, which is the saddle point of the diffusion path. The energy barrier for such a path was obtained under two different assumptions. First, by taking a frozen-lattice approximation using the HSE06 lattice parameter (equal to 4.14 Å) for the bulk-crystalline $2 \times 2 \times 2$ MgO supercell. The underlying assumption here is that the ions of the host remain fixed at their equilibrium lattice sites of the unperturbed pristine MgO crystal during the hydrogen/muonium migration. This approximation does not, therefore, consider structural relaxation effects. Alternatively, under the relaxed-lattice approximation these effects are explicitly treated by allowing the nearest neighbors of the diffusing species to displace and adjust their position during the migration process. Additional calculations by allowing all atoms to relax led to very small modifications of the migration barrier ($\approx$ 0.02 eV), suggesting that the barrier magnitude is mainly controlled by the nearest-neighbor interactions.

The results show a strong effect of structural relaxation on the barrier height (see Fig. 6). A rather high barrier of 1.83 eV is observed in the frozen-lattice approximation. This is due to strong overlap interactions between the diffusing hydrogen/muonium and its neighbors; at the saddle point (the top of the barrier), the two nearest Mg and O neighbors are 1.46 Å apart from the muon. Taking relaxation effects into account, the migration barrier drops significantly to 0.64 eV, with the four nearest neighbors moving away from the muon and eventually reaching a distance of 1.63 Å.

The barrier profiles shown in Fig. 6 are without consideration of the zero-point energy of the particle. A change in barrier height can occur if the zero-point energy is different in the ground state and in the saddle-point configuration. This can have an appreciable effect for the relatively light muonium particle. The zero-point energy correction to the migration barriers was determined here by calculating the local vibrational contributions of the muon particle [70]. These calculations were performed within a harmonic approximation for the muon motion which should be sufficient for obtaining the migration barriers at the low temperatures (< 300 K) examined in the present study. Our calculations show that zero-point effects are negligible in the frozen-lattice results because the respective sums of the vibra-

tional frequencies in the two sites cancel. However, for the relaxed geometry, a softening of the vibration frequencies by about 35% at the saddle point leads to a considerable reduction of the migration barrier of muonium by 190 meV, lowering the final barrier to 0.45 eV. Overall, the final theoretical barrier is in acceptable agreement with the experimentally extracted activation energy of site conversion (236 meV) (see section 2).

## 3.4 The doorway model

We develop here a model for the last stage of the thermalization of the implanted muon into the lattice. Since the whole process begins with a specific reaction through which the final embedding process proceeds, we call it doorway model.The "doorway state" concept has been used extensively in nuclear physics to describe intermediate structures in nuclear reactions [71]. The model has also been applied in other connections, e.g. in fullerene research, where the excitation of vibrational modes occurs via the primary excitation of a particular oscillation [72].

The muon leaves the charge exchange regime as neutral muonium $Mu^0$ or as positive charged $Mu^+$ [25]. The formation of $Mu^-$ is very unlikely at this stage. The kinetic energy of the muon after the charge exchange regime is in the order of 1 to several eV, related to the band gap of the host material [25]. The present experimental observations of a large relaxation and frequency-shift of the diamagnetic-like signal in Fig. 2 reveal the paramagnetic character of the state. This clearly indicates that, at the end of the charge-exchange process, the energetic neutral muonium $Mu^0$ configuration is present in this case. We will therefore now discuss the reaction of the neutral muonium fraction with the lattice.

After the charge exchange stage, the neutral muonium moves with the remaining kinetic energy from interstitial site to interstitial site across a potential barrier. Initially, the site changes are so fast that the lattice atoms cannot react and the muonium diffuses in the pristine (non-relaxed) lattice. The muonium loses kinetic energy through elastic scattering at the host

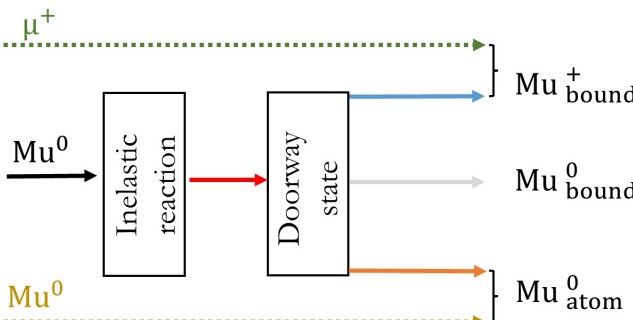

Figure 7: Flow diagram of the reaction of the muon with the host lattice at the final stage of thermalization. The middle part of the figure (solid lines) sketches the doorway reaction mechanism: incoming neutral muonium is stopped by the excitation of a local vibration of the surrounding atoms (inelastic reaction). The resulting composite configuration, the doorway state, lasts for some time (ns to $\mu$s) before decaying into either atom-like neutral muonium ($Mu^0_{atom}$) or a positively charged bound state ($Mu^+_{bound}$). In general, a neutral bound configuration ($Mu^0_{bound}$) can also be formed, but this is not the case here. There are two side reactions (indicated by dotted lines): $\mu^+$ at the end of the charge-exchange regime may react directly with the host atoms forming $Mu^+_{bound}$. Or neutral muonium can be stopped by elastic scattering only, without the strong inelastic process, forming directly $Mu^0_{atom}$.

atoms until the energy becomes so low that the muonium barely reaches the barrier height. In this case, the residence time at the top of the barrier is long enough to allow a strong inelastic reaction with the host atoms. More details will be discussed below. The proposed mechanism for the muon reaction with the host lattice is sketched in Fig. 7.

The middle part of Fig. 7 shows the doorway mechanism with the inelastic reaction and the formation and decay of the doorway state. This part will be discussed in detail below. But we would like to mention before that two other embedding reactions are possible. It is possible that at the end of the charge-exchange regime, $\mu^+$ is integrated directly into the lattice (top dashed line in Fig. 7). Another possible reaction could be that the neutral energetic muonium at the end of the charge-exchange regime is stopped by elastic scattering only and passes directly to the atomic ground state (bottom dashed line in Fig. 7). In the present case of MgO, these competing reactions seem to be of minor importance, but in other semiconductors and insulators they may be significant.

**Inelastic reaction.** The inelastic reaction is caused by the force exerted by muonium on the surrounding atoms. A rough estimate that the reaction can take place is that the residence time $\tau$ of the muonium at the top of the barrier is of the order of the inverse phonon frequency $\nu$: $\tau \approx 1/\nu$. If a frequency at the upper end of the phonon spectrum [62] is assumed for this local vibration ($\nu \approx 10^{13}$ s$^{-1}$), this results in a value in the order of $10^{-13}$ s for the dwell time of the muonium at the top of the barrier. This condition is fulfilled if the muonium has no more than a few meV kinetic energy over a length of about the Bohr radius of the muonium (0.054 nm). Of course, these classical considerations give only order of magnitude values.

**The doorway state.** Immediately after the inelastic process, the muon is still at the position it was before the reaction, i.e. in the center of the cube surface (Fig. 6), but the neighboring lattice atoms are now vibrating. We call this strongly excited phase, which lasts only nanoseconds or less, the "thermal spike" regime. Due to the coupling of the initial excitation to other phonon modes, the spike energy diffuses rapidly into the lattice and a relaxed configuration is formed [32]. This situation lasts up to hundreds of nanoseconds long and is thus observable on the timescale of $\mu$SR. The muonium state in the hot phase ("thermal spike" regime) and the subsequent relaxed configuration constitute what we call the "doorway state". We therefore define the doorway state as comprising the two phases, the short-lived hot phase and the longer-lived relaxed phase. This latter state is accessible to muon spectroscopy.

## 4 Discussion

### 4.1 Formation and decay of the doorway state

**Formation of the doorway state.** Immediately after the inelastic reaction, the muonium is in a agitated environment. In this early phase, reactions to the final states can already take place, and only a part remains in the doorway configuration, now with relaxed surroundings. We use the term "long-lived doorway state" for this longer-lived part, bearing in mind that there is a short-lived precursor hot stage in which reactions to the final states can already take place.

The doorway state lasts for hundreds of nanoseconds and is thus accessible for $\mu$SR spectroscopy. The lifetime of the long-lived state can not be determined precisely here because we can not separate exactly the slightly paramagnetic doorway state from the fully diamagnetic state which is the final state arising from the conversion. The fact that the paramagnetic interaction is seen in $\mu$SR indicates that the doorway state exists at least some time during

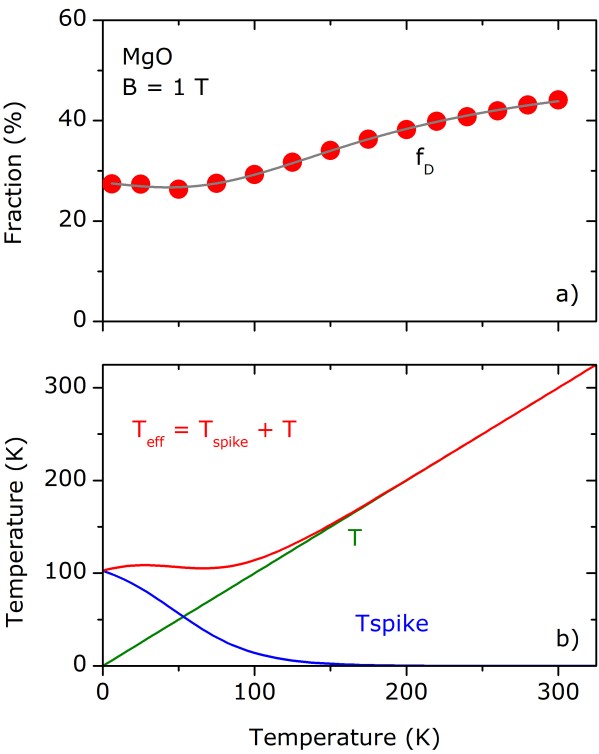

Figure 8: a) Diamagnetic-like fraction (points) and fit with Eq. 4 (solid line). b) Effective temperature $T_{\text{eff}}$ composed of the thermal spike temperature $T_{\text{spike}}$ and lattice temperature $T$.

the lifetime of the muon (2.2 $\mu s$). A similar case is the fast relaxing signal in $Al_2O_3$ [37, 73]. There, lifetimes (inverse conversion rates) of about 1 $\mu s$ were observed at low temperatures. The lifetime decreases rapidly above about 40 K and reaches a value of about 5 ns at about room temperature. A similar behaviour is expected here, but the actual values depend on the material.

We identify the long-lived doorway state with the fast-relaxing part of the diamagnetic-like signal in the experiment (shown in Fig. 2 and repeated in Fig. 8 a)). As mentioned before, the diamagnetic-like signal contains also a contribution from the purely diamagnetic $Mu^+$ configuration which cannot be separated in the present experiment. This has to be kept in mind when discussing the final results. The diamagnetic-like fraction will now be analyzed in detail.

The increase in the diamagnetic-like fraction in Fig. 8 a) above 100 K indicates that the formation of this configuration needs thermal activation. The fact that the fraction does not decrease further below 100 K is attributed to the thermal spike, which has a similar effect as an increased temperature. For the thermal spike temperature, $T_{\text{spike}}$, we assume an inverse $S-$shape temperature dependence with parameters $T_0$, $T_{1/2}$ and $k$ [32]:

$$T_{\text{spike}} = \frac{T_0}{1 + \exp\left[k\left(T - T_{1/2}\right)\right]}. \tag{4}$$

We then define an effective temperature $T_{\text{eff}} = T_{\text{spike}} + T$ and describe the diamagnetic-like fraction $f_D$ by:

$$f_D = f_0 \frac{N \exp\left(-\frac{E_a}{k_B T_{\text{eff}}}\right)}{1 + N \exp\left(-\frac{E_a}{k_B T_{\text{eff}}}\right)}, \tag{5}$$

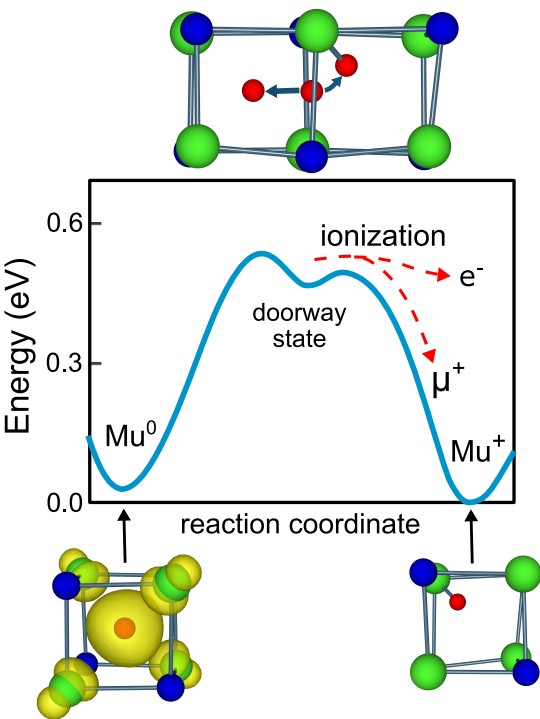

Figure 9: Schematic potential profile for muonium in the long-lived doorway state, in MgO. The decay of the long-lived doorway state occurs either to interstitial muonium, $Mu^0$, or to a bound configuration in which the muon is bound to an oxygen atom in the diamagnetic $Mu^+$ configuration and the electron $e^-$ is in the conduction band (CB). The left side of the potential curve and the energy scale are taken from the potential obtained for muonium diffusion in the relaxed lattice (red curve in Fig. 6), the right side is only schematic. The symbols indicate: green O, blue Mg, red $\mu^+$ and yellow the electron cloud.

where $E_a$ is the activation energy, $N$ the statistical weight factor and $f_0$ the total fraction. The fit result is shown in Fig. 8 a) as a solid line. The effective temperature $T_{\mathrm{eff}}$ is represented in Fig. 8 b), together with $T_{\mathrm{spike}}$ and $T$. The few experimental points in the low temperature range do not allow a clear separation between the thermal spike and the lattice temperature contribution. Since the experimental diamagnetic fraction is approximately constant below about 100 K, we have adjusted the spike temperature parameters to obtain an approximately constant effective temperature below 100 K. The fit was then performed with fixed parameters for $T_{\mathrm{spike}}$: $T_0 = 120$ K, $T_{1/2} = 47$ K and $k = 0.038$ $\mathrm{K}^{-1}$. The parameters of the thermal spike are effective values simulating the muonium reaction in the hot phase. $T_0$ is approximately the effective temperature at the muon site at $T = 0$ K. The $T_{1/2}$ value of 47 K indicates that the thermal spike effect gradually disappears around this temperature; $k$ describes the slope of the decrease with temperature.

In the final fit, $f_0$ was fixed to 78%, the value obtained in the low field experiment at higher temperatures.[1] The results yield $E_a = 12(1)$ meV and $N = 2.0(1)$. Thus there is a small barrier for the formation of the diamagnetic-like configuration but the higher statistical weight ($N \approx 2$) causes that diamagnetic-like fraction to increase with increasing temperature.

**Decay of the long-lived doorway state.** The decay of the long-lived doorway state is schematically sketched in Fig. 9. It decays either to atom-like interstitial muonium $Mu^0$ or

to a bound configuration where, in the present case, the muon is bound to an oxygen atom in a diamagnetic $Mu^+$ configuration and the electron is in the conduction band. The sketch at the top of Fig. 9 indicates the site change of the muon from the middle of the square plane to either the center of the cube or to a bound configuration with an oxygen atom. The bottom pictures in Fig. 9 display the finally formed configurations: atom-like muonium $Mu^0$ with the electron distribution around it and the diamagnetic $Mu^+$ bound to an oxygen atom.

The long-lived doorway state decays mainly by the loss of the electron. In the present case, the electron distribution appears to spread out more and more with increasing temperature, reducing the hyperfine interaction until the interaction disappears completely and a purely diamagnetic state is formed.

In principle, the long-lived doorway state could also decay to atomic muonium (see Fig. 9). However, the observed muonium lines cannot stem from this decay since dephasing due to the transition from the diamagnetic-like frequency to the muonium frequencies and the long conversion time (hundreds of nanoseconds) would destroy the phase coherence completely. But some muonium fraction from this decay may be contained in the missing fraction (see Fig. 4). The main part of the observed muonium is most likely formed in the short-lived (nanoseconds or less) thermal spike phase.

The missing fraction has two contributions: one refers to the polarization loss in the hot phase directly after the stopping of the muon. It accounts for the fraction loss from 100% to 78% fraction which is seen in the low field experiment where no phase shift is observed. The second part is attributed to dephasing in the transition of the intermediate state to the interstitial atomic configuration. It accounts for the fraction loss between 78% and the total fraction.

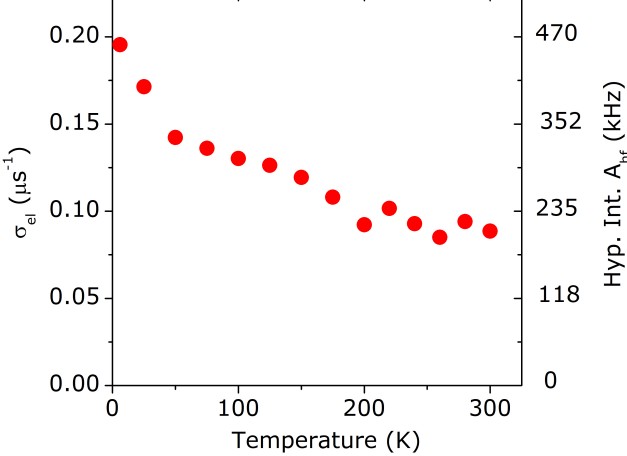

Figure 10: Electronic part of the relaxation $\sigma_{el}$ of the diamagnetic-like component, obtained by subtracting the nuclear contribution. The right scale shows the hyperfine interaction derived from the linewidth via $A_{hf} \approx \text{FWHM} \approx 2.35\,\sigma_{el}$. We attribute the electronic part of the relaxation rate to the presence of an unresolved hyperfine interaction $A_{hf}$.

## 4.2 Properties of the long-lived doorway state

**Paramagnetic interaction I: Hyperfine interaction derived from the line width**

The measured relaxation rate (linewidth in frequency space) is larger than expected if only nuclear moments were causing the line broadening. To obtain the electronic contribution alone (Fig. 10), the nuclear contribution was subtracted via $\sigma_{\text{el}} = \sqrt{\sigma_D^2 - \sigma_{\text{nuc}}^2}$. Here $\sigma_D$ is the measured diamagnetic-like relaxation and $\sigma_{\text{nuc}}$ the nuclear contribution which was calculated [59, 60] for the muon configuration shown in Fig. 5 c). The calculated value of the nuclear interaction is $\sigma_{\text{nuc}} = 0.02 \, \mu\text{s}^{-1}$.

The hyperfine interaction obtained in this way has values on the order of a few 100 kHz (right scale in Fig. 10). These very small values (about five orders of magnitude smaller than those of the interstitial muonium) indicate that the electron is widely distributed and has very little overlap with the muon.

The hyperfine interaction (right scale in Fig. 10) decreases continuously with increasing temperature from about 450 kHz to about 120 kHz. One reason for this is electron spin fluctuation, which changes the sign of the interaction leading to a narrowing of the line. Another possibility is that the electron moves further outward with increasing temperature, reducing the residence time at the muon site. Eventually, the electron may be lost completely, leaving $\text{Mu}^+$ behind. In the present case of MgO, it is not possible experimentally to separate the slightly paramagnetic and the purely diamagnetic state.

**Paramagnetic interaction II: Frequency shift at high field ($B = 1$ T)**

The frequency shift at high fields (Fig. 2 b)) is caused by the polarization of the electron spin and is large only at high fields and low temperatures. For unresolved hyperfine lines, we can assume that the frequency shift $\Delta \nu$ is given by the average frequency of the two lines weighted by their intensities. The line intensities are given by the polarization of the electron spin $P_e(B, T)$, which depends on the field $B$ and the temperature $T$. Assuming thermodynamic equilibrium for the electron spin, one obtains

$$\Delta \nu = \frac{A_{\text{hf}}}{2} P_e(B, T) = \frac{A_{\text{hf}}}{2} \tanh\left(\frac{\mu_B B}{k_B T}\right). \tag{6}$$

Here $A_{hf}$ is the hyperfine interaction, $\mu_B$ the Bohr magneton and $k_B$ the Boltzmann constant. With the approximation $\tanh x \approx x$ for small $x$, one obtains:

$$\frac{\Delta \nu}{\nu} = \frac{\gamma_e}{\gamma_\mu} \frac{h A_{\text{hf}}}{4 k_B T} = \left(248.6 \times 10^{-11} \, \text{K.s}\right) \frac{A_{\text{hf}} \left(\text{s}^{-1}\right)}{T(\text{K})},$$

$$A_{\text{hf}} \left(\text{s}^{-1}\right) = 0.4 \times 10^9 \frac{\Delta \nu}{\nu} T(\text{K}). \tag{7}$$

Fig. 11 shows the relative frequency shift $\Delta \nu / \nu$, normalized to the Ag calibration, on the left scale, and $\Delta \nu' / \nu$ normalized to the asymptotic behaviour at high temperatures, at the right scale (see the following discussion). Since the shifts are very small (in the ppm range), a close look at the errors is appropriate: the statistical error of the frequency measurement is about $1-2$ ppm. The temperature dependence measurements between 25 K and 300 K were made in a single run without changing the field. The stability of the magnet is in the ppm range, so the magnet instability does not contribute to the error. The 6 K point was measured in a different run. Since the reproducibility of the field setting may have a large error, we did not include this point in the fitting procedure, but included the point in the final plot.

We first fitted the data normalized to the Ag calibration (left scale in Fig. 11) with a $1/T$ dependence (Eq. 7), but allowed a constant offset. The result yielded an offset of -14(5) ppm.

On the right scale of Fig. 11 we represent the relative shift $\Delta\nu'/\nu$, corrected for this fitted offset. This corresponds to a normalization of the frequency shift to the asymptotic behavior at high temperatures where no shift is expected. For the data between 25 K and 300 K, we assumed an error of 5 ppm (corresponding to the uncertainty in the offset), whereas for the 6 K point an estimated error of 200 ppm was adopted.

As can be seen in Fig. 11, the frequency shift is negative. Since the electron polarization is positive, the hyperfine interaction must be negative. Negative hyperfine interactions are not uncommon for bonded muonium configurations, as observed, for example, for the contact term of bond-centered muonium in group IV elements [29].

The hyperfine interaction derived from the corrected $\Delta\nu'/\nu$ frequency shift is of the order of $1-2$ MHz in the low-temperature range (see insert of Fig. 11). At higher temperatures, the errors are very large and no reliable values of hyperfine interaction can be extracted. The value obtained at low temperatures is much larger than the hyperfine interaction obtained from the linewidth (Fig. 10), where values of the order of a few 100 kHz were measured. This apparent discrepancy can be explained as follows: i) the linewidth measures the *average* value of the hyperfine splitting. Electron spin fluctuations narrow the splitting. ii) On the other hand, the frequency shift measures the *position* of the average value, which depends on the intensity of the two hyperfine lines. The fluctuations keep (or even establish if not already present) the thermodynamical equilibrium occupation of the hyperfine lines.

Thus, the measured frequency shift gives the *actual* hyperfine splitting, while the linewidth yields the *average* value for the fluctuating electron spin. The hyperfine interaction displayed in the inset to Fig. 11 shows some variation with temperature. These could be real effects, but the errors are large, so we do not discuss this further. We note that the fit in Fig. 11 clearly only describes the gross variation of the relative frequency shift and that other minor effects may be playing a role. The present data do not allow however for a more detailed analysis.

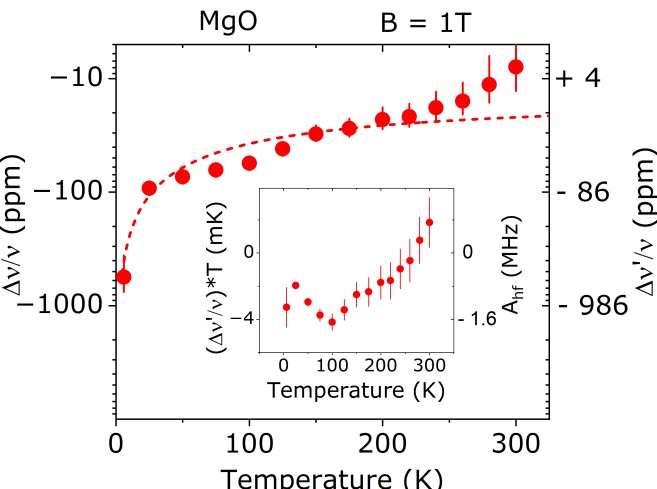

Figure 11: Relative frequency shift at $B = 1$ T as a function of temperature: the left scale shows $\Delta\nu/\nu$ normalized to the Ag calibration, the right scale shows $\Delta\nu'/\nu$ normalized to the asymptotic behavior, assuming that the shift is zero at high temperatures. The dashed line is a fit with a $1/T$ dependence (Eq. 7). The insert shows $\left(\Delta\nu'/\nu\right)\times T$ on the left and the corresponding hyperfine interaction $A_{hf}$ on the right.

## 5 Conclusions

In the present study, combining $\mu$SR measurements and DFT calculations, detailed information about the reaction of muonium with the host lattice in MgO is obtained. The experimental data are well described by a model that we call "doorway model". The basic assumption of this model is that neutral muonium is stopped at the end of the implantation path by a strong inelastic process. The inelastic reaction occurs at the top of the diffusion barrier, where the muonium is slow enough to excite a local vibration. This excitation is not caused by the impact of the moving particle, but by the expansion force of muonium. After this reaction, the muon is stopped, and a compound system is formed which decays independently from its formation. First a short-lived hot phase and subsequently a longer-lived relaxed phase are formed. This latter state ("long-lived doorway state") is accessible to muon spectroscopy.

The long-lived doorway state is characterized by a widespread electron distribution around the oxygen-bound muon with an average hyperfine interaction in MgO on the order of a few 100 kHz. The momentary hyperfine interaction is in the order of 1 to 2 MHz but is reduced to a smaller average value by electron spin fluctuation. The presence of the electron in the diamagnetic-like long-lived doorway state is evidenced by the line broadening and by the frequency shift of the diamagnetic-like line. The negative value of the paramagnetic shift indicates that the sign of the hyperfine interaction is negative. The paramagnetic interaction is observed up to temperatures above 300 K, suggesting that the electron remains bound to the muon in MgO beyond 300 K.

MgO is a favorable case for these investigations due to its simple lattice structure and low density of nuclear magnetic moments, allowing a clear observation of the electronic interaction. We expect that the doorway model can be applied, at least in part, to $\mu$SR studies of other semiconductors and insulators.

## Acknowledgments

Muon beam time allocation from the Laboratory for Muon-Spin Spectroscopy at the Paul Scherrer Institut and the support of the PSI muon team are gratefully acknowledged. Technical assistance from Dr. Tatsuo Goko is particularly acknowledged. The authors also acknowledge the use of the computing facilities of CFisUC and the Department of Physics of the University of Coimbra. A. W. thanks Prof. Klaus Lips for support of this work.

**Funding information** This work was financed through national funds by FCT - Fundação para a Ciência e Tecnologia, I.P. in the framework of the projects UIDB/04564/2020 and UIDP/04564/2020, with DOI identifiers 10.54499/UIDB/04564/2020 and 10.54499/UIDP/04564/2020, respectively.

## A Methods

### A.1 Experimental details and methods

The sample was a commercial high-purity ($> 99.99\%$) MgO single crystalline substrate from Alfa Aesar, with dimensions $10 \times 10 \times 1$ mm$^3$. The crystallographic orientation of the sample was such that one of the axes pointed in the beam direction, and the other two axes were perpendicular to it in the horizontal and vertical direction. This was checked by X-ray diffraction, which also confirmed the high crystalline quality of the sample.

The $\mu$SR experiment was performed at the Swiss Muon Source of the Paul Scherrer Institut in Switzerland [57] with the high-field spectrometer (HAL-9500) [58] in transverse geometry. A magnetic field $B = 1$ T was used. The magnetic field was applied parallel to the beam direction which means, considering the crystal orientation mentioned above, parallel to a cube axis of the fcc structure. The muon spin was rotated, before the implantation, from its original direction (antiparallel to the beam) towards the vertical direction to allow the transverse field measurement.

The muons are implanted into the sample with energy of about 4 MeV and stop well inside the sample (a few hundred $\mu$m below the surface). The muon spin rotation ($\mu$SR) is measured by detecting the anisotropic emission of positrons as a function of time [17, 18]. The experimental arrangement of the HAL-9500 spectrometer consisted of eight forward and eight backward positron detectors, arranged in rings around the muon beam. Data analysis was performed using the Musrfit [74] and WiMDA [75] software.

Figure 1 shows the Fourier spectrum of $\mu$SR data at T = 6 K and B = 1 T. Three lines, a diamagnetic-like line and the two components $\nu_{12}$ and $\nu_{34}$ of muonium are seen. The data were therefore analysed with three relaxing components, as follows

$$
\begin{aligned}
A(t) = {} & A_D \exp\left(-\frac{1}{2}\sigma_D^2 t^2\right)\cos\left(2\pi\nu_D t + \phi_D\right) \\
& + A_{12}\exp\left(-\lambda_{12}t\right)\cos\left(2\pi\nu_{12}t + \phi_{12}\right) \\
& + A_{34}\exp\left(-\lambda_{34}t\right)\cos\left(2\pi\nu_{34}t + \phi_{34}\right),
\end{aligned}
\tag{A.1}
$$

where $A_i$, $\nu_i$, $\phi_i$ represent the corresponding amplitude of the oscillation, precession frequency and initial phase, respectively; the index $i = D, 12$ or $34$ indicates the diamagnetic-like line, the muonium 12 line and the muonium 34 line, respectively; $\lambda_i$ and $\sigma_i$ represent the spin relaxation rates, which have been found in a preliminary analysis to be better described by a lorentzian shape for the muonium lines and by a gaussian shape for the diamagnetic line. A global fit was performed for all sixteen detectors of the HAL9500 instrument, where the number of positron counts in each detector was fitted to $N_i(t) = N_i^0 \exp\left(-t/\tau_\mu\right)[1 + A_i(t)]$, with $A_i(t)$ given by Eq. A.1, $\tau_\mu$ the muon lifetime and $N_i^0$ the normalized number of counts for each detector. Thereby the frequency and relaxation rate were assumed to be the same for all detectors, but the phases were left free. The fractions of muons forming each state were obtained by comparing with the maximum instrumental asymmetry $A_{\mathrm{max}}$ obtained from a calibration measurement with silver.

As mentioned above, the eight forward as well as the eight backward detectors were arranged on a ring around the beam direction. The nominal rotation angle between two adjacent forward or backward detectors is thus approximately $45°$. The fits with individual phases for each detector gave indeed a phase difference between two adjacent detectors of $45°$ within $1°$ variation. We therefore performed also fits with fixed relative phases of $45°$, and a phase difference between a reference forward detetor and a reference backward detector. These fits gave the same chi-squares as the fits with free phases. Therefore, the final analysis was performed with fixed relative phases with respect to the phase of the reference detectors. By this procedure only a single effective phase is obtained for each frequency.

### A.1.1 Correction for time resolution

For high-frequency lines, the finite time resolution reduces the observed amplitude. A gaussian dependence on the frequency $\nu$ is assumed for the correction function $C(\nu)$:

$$
C(\nu) = \exp\left[-(2\pi\nu\sigma)^2/2\right].
\tag{A.2}
$$

For the width $\sigma$, the value $\sigma = 86$ ps from Ref. 58 was adopted. All amplitudes, including those of the Ag calibration, were corrected for this effect. For the diamagnetic component $f_D$,

the correction due to the time resolution has no effect, but the muonium lines are strongly affected. The uncertainty due to the correction is shown as an error bar in Fig. 4.

## A.2 DFT calculations: Setup and preliminaries

The DFT calculations were carried out by the *ab-initio* VASP code [64–66]. The corresponding implementation is based on the projector augmented-wave method [76, 77] and the use of pseudopotentials to represent the core-valence interactions. Accordingly, the valence-electron wavefunctions were expanded by taking a plane-wave basis limited by a cutoff of 440 eV. Exchange and correlation effects between the electrons were described by the semilocal PBE functional [67] as well as by the HSE06 hybrid-functional approach which includes a fraction of 35% of exact non-local exchange [68,69]. The hyperfine constants for the neutral hydrogen (muonium) states were determined by adopting the approach in Ref. [78].

Whereas the PBE functional gives a very low energy band gap (equal to 4.98 eV), the HSE06 approach instead yields a gap equal to 7.92 eV in excellent agreement with the reported experimental gap, 7.8 eV, of MgO [56].

For the defect calculations the hydrogen/muon particle was treated as an impurity and was embedded inside $2\times2\times2$ (64−atom) bulk-crystalline MgO supercells. Minimization of the total zero-temperature internal energies led to the formation energies and charge-transition levels for all possible hydrogen/muonium configurations [79]. This treatment should be sufficiently accurate for low temperatures (less than 300 K) where the present $\mu$SR measurements were performed with anharmonic effects expected to be negligible [80]. The final results of these quantities reported in the present study were obtained by means of the HSE06 functional.

Migration-energy profiles were determined by means of a constrained-path approach [81]. The energy barriers were obtained both for the frozen lattice and also accounting for structural relaxation along the diffusion paths (relaxed lattice).

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
