# Peer review of "Muonium reaction in MgO: A showcase for the final steps of ion implantation"

_SciPost Physics, doi:SciPost Phys. Core 8, 056 (2025)_

## Round 1 · Referee Report · Anonymous (Referee 1) · 2025-5-25

Report

This is an interesting manuscript in which the authors elaborate on their earlier transition state model. In the muSR community, the formation of the final muon/muonium configuration in a material is still a matter of debate (see refs. [27] and [28]), and here the authors provide additional evidence for their model, where some fraction of the muons - before coming to its final configuration - go through a transition state with a weak hyperfine coupling to an unpaired electron. This explains the appearing enhanced depolarization rate and observed frequency shift of the diamagnetic(-like) line in the muSR spectra. These observations cannot be explained by the alternative model: here, neutral muonium formation is assumed to take place by a delayed capture of an electron by the stopped muon where the elecron comes from the ionization track of the muon. While the author's model cannot exclude that this delayed process could happen for some fraction of the formed neutral muonium atoms, it naturally explains the observed effects on depolariation rate and frequency shift of the diamagnetic-like signal. The authors' interpretation is further substantiated by DFT calculations. Overall, I find the manuscript well and clearly written, and I recommend publication after the authors addressed the points below.

Requested changes

1- Why do the authors call their suggested intermediate muonium configuration "doorway state", and no longer "transition state"? To me, that sounds more like a linguistic quibble, but perhaps I missed something. In other words, what is the difference between the doorway and the tranistion state model of ref. [31]? Is it the "inelastic reaction" used in the doorway state? But this looks more like adding more details to what is happening during the time the muon spends in the transition state.

2- what is the origin of the missing fraction in Fig. 4? The authors explain it only later in the discussion in chapter 4. I would prefer to get already an indication about the origin of the missing fraction in the description of Fig. 4.

3- In lines 163/164, the authors write that the "apparent peak in the amplitudes [of the Mu lines] at around T = 100 K will be discussed in connection with Fig. 4". However, there is no discussion about this peak in connection with Fig. 4 on page 7. I am curious to see the interpretation of the authors of this peak.

4- what is the parameter C in Eq. 3?

5- ref. [24]: Advances in Physics 72(4), 409 (2023)

6- ref. [58] does not seem to be the proper reference for the Swiss Muon Source. Ref. [58] refers to one of the beamlines of the SmuS.

Recommendation

Ask for minor revision

  • validity: top
  • significance: good
  • originality: top
  • clarity: high
  • formatting: excellent
  • grammar: excellent

Author:  Rui Vilão  on 2025-06-12  [id 5565]

(in reply to Report 1 on 2025-05-25)
Category:
answer to question

We thank the referee for the careful review of the paper and the comments. We now reply to the comments and indicate our proposed changes:

  1. “Doorway model” expresses more clearly than “transition state model” the entrance character of the initial reactions: The passage through the “doorway” by the inelastic reaction and the formation of an initial configuration through which all other reactions proceed. We will add a corresponding sentence in the paper.

  2. We will add in the description of Fig. 4 in the main manuscript: “About 20 % fraction is missing. It is attributed to muon spin polarization loss due to rapid fluctuations of the hyperfine interaction in the initial hot phase after the muon stopping.”

  3. We will add the following in the manuscript when introducing Fig. 4, before the last paragraph of section 2: “As mentioned in the experimental section, the observed muonium amplitudes show a peaking slightly above 100 K. We note that the increase is parallel to the increase of diamagnetic-like fraction, suggesting that both states are formed in the same process. The decrease of the muonium fraction at temperatures above about 120 K may either be due to a decrease of the formation probability or to dephasing effects which are rather strong at these high frequencies.”

4 – C is a constant fitting parameter. It is related to the coupling strength of the muonium electron with the phonons, but no details can be given here. We will add this in the main text.

5 – We will correct this and apologize for the wrong citation.

6 – We will replace this reference by the PSI webpage.

---

## Round 1 · Referee Report · Anonymous (Referee 2) · 2025-6-8

Report

The manuscript presents muon spin spectroscopy experiments in conjunction with DFT calculations to explore muon thermalization in MgO. A central aspect of the work is the proposal of a doorway model, wherein the muon is hypothesized to pass through a metastable intermediate state before reaching its final embedding site(s). The manuscript is generally well-written, and the integration of high-field muon measurements with theoretical site calculations in MgO represents a novel approach that could contribute meaningfully to the existing body of literature.
That said, the main objective of the study is the introduction and validation of the doorway model would benefit from a more rigorous and comprehensive foundation. At present, the model's description appears to rely on assumptions that are not sufficiently supported by the presented experimental data or theoretical analysis. In particular, the physical mechanisms and associated timescales underlying the model remain speculative, and the current first-principles calculations do not fully substantiate the proposed conclusions or their quantitative alignment with the experimental observations.
While the study explores an interesting and potentially impactful idea, I believe that further experimental validation and more detailed theoretical modeling are necessary to support the claims made. I encourage the authors to address these issues in a revised version, which could significantly strengthen the manuscript. However, in its current form, I do not recommend the manuscript for publication.

The concern is;
There is insufficient and unambiguous evidence to support the claim that the transient state proposed in the model is long-lived. Only a qualitative experimental estimate was provided. Furthermore, there is a lack of compelling evidence for the values of the fixed parameters used in fitting the thermal spike. The authors have also reported significant uncertainties in their analysis of the various phases. In addition, no definitive proof has been presented to demonstrate that the muon is halted at the transient state due to an inelastic reaction. The evidence provided relies primarily on classical, assumed estimates of phonon-induced charge density fluctuations, with no direct simulations to substantiate the muonium effects.
Given that the authors have already utilized first-principles calculations to determine muon localization sites, and considering the novel nature of the proposed processes, it would be beneficial for the authors to undertake the challenging task of employing appropriate methodologies for studying dynamic processes. This would not only help validate both the short-lived and long-lived processes proposed, but also provide stronger support for the model's overall validity.

Recommendation

Ask for major revision

  • validity: low
  • significance: ok
  • originality: good
  • clarity: high
  • formatting: excellent
  • grammar: excellent

Author:  Rui Vilão  on 2025-06-17  [id 5577]

(in reply to Report 2 on 2025-06-08)
Category:
reply to objection

We thank the referee for reviewing this manuscript and for his comments, but we disagree with his/her assessment. We believe that the experimental data presented in this work, together with joint calculations, constitute a solid, serious, novel and impactful contribution to the field of muon/ion implantation in solids. This paper is a contribution to the discussion, but it is not meant as the final solution of the problem. We kindly ask the referee to reconsider his/her assessment and to agree to acceptance of the paper.

We now reply to specific comments of the referee:

Comment of referee 2
There is insufficient and unambiguous evidence to support the claim that the transient state proposed in the model is long-lived. Only a qualitative experimental estimate was provided.

Our comment
The fast component is long-lived, otherwise it would not be experimentally observed. A precise determination of the lifetime is out of the scope of this work, but the experimental evidence is unambiguous.

Comment of referee 2
Furthermore, there is a lack of compelling evidence for the values of the fixed parameters used in fitting the thermal spike.

Our comment
The thermal spike concept is used here only to account for the levelling-off of the formation probability at low temperatures. It is not a major issue in this connection and is therefore a secondary point that does not affect the main model.

Comment of referee 2
The authors have also reported significant uncertainties in their analysis of the various phases.

Our comment
The determination of the phases is very uncertain for high frequency signals. We therefore do not discuss the phases explicitly in this paper.

Comment of referee 2
In addition, no definitive proof has been presented to demonstrate that the muon is halted at the transient state due to an inelastic reaction. The evidence provided relies primarily on classical, assumed estimates of phonon-induced charge density fluctuations, with no direct simulations to substantiate the muonium effects.

Our comment
For an interaction of the muon with the lattice it is necessary that the muon stays a certain time at a place. This requirement is fulfilled during the slowing down, first at the top of the diffusion barrier. We give a classical consideration at what energy the sufficiently long stay time is reached, but this is of course only a qualitative estimate.

Comment of referee 2
Given that the authors have already utilized first-principles calculations to determine muon localization sites, and considering the novel nature of the proposed processes, it would be beneficial for the authors to undertake the challenging task of employing appropriate methodologies for studying dynamic processes. This would not only help validate both the short-lived and long-lived processes proposed, but also provide stronger support for the model's overall validity.

Our comment
Simulations such as those asked by the referee are out of the scope of the present paper and would represent alone a major contribution to the difficult field of simulation of the final stages in ion implantation physics.

---

## Round 2 · Referee Report · Anonymous (Referee 1) · 2025-6-30

Report

The authors addressed all questions well, and I recommend publication after the authors corrected the following problem with the references:

  • In line 534, ref. [76] should be replaced by the ref. [58]. There is no relation of ref. [76] (this was ref. [58] in version 1 of the manuscript) to the paper and it can be removed.

Requested changes

  • In line 534, ref. [76] should be replaced by the ref. [58]. There is no relation of ref. [76] (this was ref. [58] in version 1 of the manuscript) to the paper and it can be removed.

Recommendation

Ask for minor revision

  • validity: high
  • significance: high
  • originality: high
  • clarity: top
  • formatting: excellent
  • grammar: excellent

Author:  Rui Vilão  on 2025-07-23  [id 5670]

(in reply to Report 1 on 2025-06-30)
Category:
correction

Sorry for the wrong citation. We have now corrected it.

---

## Round 2 · Referee Report · Jess Brewer (Referee 3) · 2025-7-4

Strengths

See attached PDF.

Weaknesses

See attached PDF.

Report

See attached PDF.

Requested changes

See attached PDF.

Attachment

Recommendation

Publish (surpasses expectations and criteria for this Journal; among top 10%)

  • validity: good
  • significance: high
  • originality: top
  • clarity: good
  • formatting: excellent
  • grammar: excellent

Author:  Rui Vilão  on 2025-07-23  [id 5671]

(in reply to Report 2 by Jess Brewer on 2025-07-04)

Referee 3 (Prof. Jess Brewer)
We thank Prof. Jess Brewer for this fair, thoughtful, critical and inspiring review of our paper. We appreciate his comments and questions and answer them as follows

Referee 3
The paper begins with a list of assorted applications of energetic ions (especially protons) in which (the authors declare) the final stages of the stopping process are poorly understood, despite being essential to effective utilization. Since no description is given of what is unknown in the several cases, the reader is left to either read all the references or just take the authors’ word for it.

Our answer
We have rephrased this part as: "Although the stopping process is rather well understood for the high-energy region, very little is known about the final thermalization process [8–12]. The reason for this is that conventional experimental methods are not able to investigate this process at the atomic level. Muon spectroscopy offers the unique possibility that the final steps can be observed directly on the implanted particle and thus detailed information about these final steps can be obtained.”

Referee 3
Footnote 1: For some reason they refer to the 1947 Fermi and Teller paper concerning µ− capture.

Our answer
We wanted to express that stopping of muons was already on the agenda at the beginning of muon research. But this mentioning is not necessary here and we have skipped this reference.

Referee 3
Footnote 2: At the bottom of p. 11 they say, “The inelastic reaction is caused by the force exerted by the squeezed muonium on the surrounding atoms.” If the Mu0 is “squeezed” rather than “expanded”, then why is the hyperfine interaction weaker than in vacuum?

Our answer
We skipped the word “squeezed” here and the word “compressed” in a similar context in the conclusion section.

Referee 3
I have some problems with this picture.
First, the “doorway state” is proposed to last for “hundreds of nanoseconds” before losing its electron completely, during which time it has a slightly shifted precession frequency due to the loosely associated, thermally-polarized electron, after which it precesses at the free muon Larmor frequency ωD. If so, then a Fourier transform omitting the first few hundred ns should yield a sharp line at ωD. Was this done? If not, it should be. If so, and it yielded the same broadened and shifted line as the full time spectrum, then this picture cannot be correct.

Our answer
Analyzing the time spectra separately in the earlier and later times regime gives similar information as a two-component fit with a fast and a slow relaxation. We have tried such fits and obtained of course a fraction with slow relaxation. But the data were not good enough to allow a reliable extraction of the lifetime. We discuss that in section IV.2, third paragraph.

Referee 3
Second, the energy scales seem inconsistent. A fully-bound, compact Mu0 state is bound by some substantial fraction of a Rydberg unless its electron is already shared with the lattice. (Weakly bound Mu0 states are frequently observed in solids, but they always involve the lattice intimately.) Therefore the picture of a epithermal weakly-bound Mu0 state diffusing rapidly through the lattice is self-contradictory. Such diffusion is only possible for a compact, strongly-bound Mu0 state, which therefore arrives at the crucial inelastic collision with a significant fraction of a Rydberg of energy. This energy is supposedly released in said inelastic process, contributing to the local “thermal spike” that raises the effective temperature of the local lattice (which must dramatically reduce the thermal equilibrium polarization of the weakly-bound electron in the “doorway state”). How is it then possible that the resultant frequency shift is so strongly temperature dependent?

Our answer
The energetics discussion is based on the DFT calculations and the data shown in Fig. 6. The diffusing muonium is a compact atom similar to ground state muonium at the interstitial site. At the top of the barrier, the “forward directed” kinetic energy is only a few meV or less and is fully transferred to the oscillating atoms after the inelastic reaction. The inelastic reaction and the subsequent relaxation of the lattice lead then to the long-lived doorway state. The total energy transfer to heat is about 1.2 eV (difference of the barrier heights in Fig. 6).

Referee 3
It may be that my difficulty in following the logic of this paper is due to the somewhat artificial separation of experimental observations from the theory that explains their interpretation; this is probably structurally mandatory, but I get confused when flipping back and forth between assertions and their separate justifications.

Our answer
We had the same difficulty when writing the manuscript, but found that the current structure, showing very clearly the experimental data first and then building the model, is the one which gives the reader the most clear separation of experimental data and theoretical model.

---

## Round 2 · Referee Report · Anonymous (Referee 4) · 2025-7-19

Report

See below the requested experiments.
On the basis of the requested experiments (see below) the major revision should be done.
I do not recommend publication of the current version of the manuscript.

Requested changes

The authors must set a solid experimental grounds before suggesting a model. For that: 1. The authors must carry out electric field experiments at various magnetic fields. 2. The authors must measure magnetic field dependences of the muonium amplitude and the initial phase of the muonium precession.

After that is done: 3. The suggested model should be set against different models of Mu formation, in particular, the model of delayed Mu formation via an electron capture by the muon.

Recommendation

Ask for major revision

  • validity: low
  • significance: good
  • originality: poor
  • clarity: ok
  • formatting: reasonable
  • grammar: good

Author:  Rui Vilão  on 2025-07-23  [id 5672]

(in reply to Report 3 on 2025-07-19)

Referee 4
We do not agree with the assessment of the paper of Referee 4. We think the model presented in the paper has a solid foundation by the theoretical and experimental data presented in the manuscript.

Referee 4
The authors must set a solid experimental grounds before suggesting a model. For that: 1. The authors must carry out electric field experiments at various magnetic fields. 2. The authors must measure magnetic field dependences of the muonium amplitude and the initial phase of the muonium precession.

Our answer
In the present case, electric field measurements are not necessary. Such measurements cannot distinguish between the different models because the behavior of the weakly bound electron is the same in both models. The same applies to magnetic field measurements. The justification for our interpretation is based on the broadened diamagnetic line and the frequency shift.

Referee 4
After that is done: 3. The suggested model should be set against different models of Mu formation, in particular, the model of delayed Mu formation via an electron capture by the muon.

Our answer
In the introduction, we describe the competing models and mention the relevant literature. We also mention the controversy about these models and cite the references about it. In the other sections of the paper, a comparison of the models is implicitly addressed when we talk about the broadened diamagnetic-like line and the frequency shift, which shows that the paramagnetic interaction is present from the beginning.
In our opinion, the model is sufficiently well-founded experimentally and theoretically to be presented to the scientific community for discussion.

---

## Round 2 · Author Response

We resubmit our manuscript after taking into account the comments of the referees.

---

## Round 2 · List of Changes

As a reply to the comments of Referee 1:

  1. “Doorway model” expresses more clearly than “transition state model” the entrance character of the initial reactions: The passage through the “doorway” by the inelastic reaction and the formation of an initial configuration through which all other reactions proceed. We added a corresponding sentence in the paper.

  2. We added in the description of Fig. 4 in the main manuscript: “About 20 % fraction is missing. It is attributed to muon spin polarization loss due to rapid fluctuations of the hyperfine interaction in the initial hot phase after the muon stopping.”

  3. We added the following in the manuscript when introducing Fig. 4, before the last paragraph of section 2: “As mentioned in the experimental section, the observed muonium amplitudes show a peaking slightly above 100 K. We note that the increase is parallel to the increase of diamagnetic-like fraction, suggesting that both states are formed in the same process. The decrease of the muonium fraction at temperatures above about 120 K may either be due to a decrease of the formation probability or to dephasing effects which are rather strong at these high frequencies.”

4 – C is a constant fitting parameter. It is related to the coupling strength of the muonium electron with the phonons, but no details can be given here. We added this in the main text, together with the remaining parameters of the equation.

5 – We corrected this and apologize for the wrong citation.

6 – We replaced this reference by the PSI webpage.

---

## Round 3 · Referee Report · Anonymous (Referee 4) · 2025-7-24

Report
The model itself does not stand critics expressed in Phys. Rev. B 101, 077201 (2020), in particular, regarding timescales and energies involved.
Recommendation
Reject

Anonymous on 2025-07-30 [id 5694]
The "anonymous" Referee is obviously one of my coauthors on the critique paper Phys. Rev. B 101, 077201 (2020) who is less charitable than I am regarding the intransigence of Vilao et al. My position is that the authors of the current paper should be given all the rope they need to hang themselves in public, but I have a lot of sympathy for my own coauthors: Vilao et al. have again systematically ignored the results of our electric field experiments, presumably because those results are incompatible with their "transition state" (a.k.a. "doorway state") model. I mention all this because I am sorely tempted to just agree with the other Referee and reject this paper, but I feel more obliged to acknowledge the ingenuity of their model and the detailed experiments they have done this time, and leave the ultimate evaluation to general scientific consensus, which requires publication. Ultimately, the hard decision is in the Editor's hands.

---

## Round 3 · Referee Report · Jess Brewer (Referee 3) · 2025-7-30

Strengths
- Substantial and thorough measurements of muSR spectra in the sample.
- Sophisticated use of DFT calculations to determine muon & muonium sites and lattice responses.
- Ingenious deployment of a toy model.
Weaknesses
- Electric field measurements that conflict with the model are ignored completely.
- Unshakeable faith in a toy model.
Report
Requested changes
None at this point.
Recommendation
Publish (easily meets expectations and criteria for this Journal; among top 50%)

---

## Round 3 · Author Response

List of changes

---

## Round 3 · List of Changes



---

## Editorial Decision

published